# Research of Organic Rankine Cycle Energy Characteristics at Operating Modes of Marine Diesel Engine

Sergejus Lebedevas and Tomas Čepaitis *

Marine Engineering Department, Faculty of Marine Technology and Natural Sciences, Klaipeda University, LT-91225 Klaipėda, Lithuania; sergejus.lebedevas@ku.lt
* Correspondence: tomas.cep7@gmail.com

**Abstract:** The publication examines one of the most effective ways to decarbonize marine transport, specifically the secondary heat sources utilization in the cogeneration cycle of the main engines. The research focuses on the optimization of Organic Rankine Cycle (ORC) performance parameters by combining them with the exhaust energy potential of a medium speed four-stroke main diesel engine in ISO8178 (E3) load cycle modes. Significant advantages were not found between the evaluated Wet-, Isentropic-, and Dry-type liquids (*R134a*, *R141b*, *R142b*, *R245fa*, *Isopentane*) in terms of ORC energy performance with a 10% difference. The use of a variable geometry turbogenerator turbine with Dry-type (*R134a*) working fluid is characterized by the highest ORC energy efficiency up to 15% and an increase in power plant (including turbogenerator generated mechanical) by 6.2%. For a fixed geometry turbine, a rational control strategy of the working fluid flow ($G_{d.sk} - \pi_T$) is determined by the priorities of the power plant in certain load modes. The influence of the overboard water temperature on the ORC energy indicators does not exceed $\pm 1\%$; however, it influences the thermodynamic saturation parameters of the working fluid condensation and, in connection with that, the fluid selection.

**Keywords:** organic rankine cycle; heat recovery; energy efficiency; main engine load cycle





## 1. Introduction

The problem of maritime transport decarbonization, as a component of the general problem of reducing $CO_2$ emissions in the transport sector, is being addressed on the normative basis of IMO standards [1]. In July 2011, the Marine Environment Protection Committee introduced an energy efficiency design index (EEDI) requirement implemented on January 1, 2013; this requirement will be made more stringent by three phases every five years starting from 2015 [1]. The EEDI requirement is estimated to achieve a 10–50% potential reduction of $CO_2$ emission per transport task. According to the EEDI requirement, new ship designs need to satisfy the $CO_2$ reduction level set for the majority of new ships based on the reference level for each ship type [2]. The complex EEDI calculation equation involves engine parameters and innovative technologies expressed in grams of $CO_2$ per ship capacity mile. The $CO_2$ emissions are reduced by improving the EEDI of the ship [3–5].

New restrictions of the marine transport $CO_2$ emissions and the instability of the fossil fuels reserves forced development of solutions for reduction. Development of the high efficiency combustion engines and alternative fuel, such as bio-ethanol, can be used as the burning of the biofuels does not cause an increase of the $CO_2$ in the atmosphere [6,7]. Dual fuel is another promising technology for marine diesel engines which can decrease $CO_2$ emissions up to 25% and $NO_x$ emission up to 85% due to the natural gas lower carbon to hydrogen ratio and flexible control of premixed fuel fraction, regardless of the operation conditions [8,9]. This technological direction of research, along with cogeneration of secondary heat sources of ship engines, are the main directions solving the problem of marine transport decarbonization in accordance with the current and promising EEDI requirement.

Considering $CO_2$ emission reduction potential, several recent studies [10] concluded that, among the current operational and design measures, only waste heat recovery systems can achieve a potential reduction of approximately 50% of the fuel energy wasted through exhaust gases and cooling jackets [11–14]. A considerable amount of fuel saving can be achieved in ships by utilizing the exhaust gas heat from internal combustion engines (ICE) and gas turbine units [15,16]. The promising use of cogeneration cycles in ship ICEs can help achieve an energy efficiency level of 50–55%; further reduction can be achieved by largely using organic working fluids. The exhaust gas temperatures of various types of marine engines range between 260 and 450 °C, which makes it possible to generate the necessary amount of steam for use in system boilers that would allow increasing the energy efficiency by up to 10% and satisfy the heat and electricity requirements of household consumers. Further, the temperatures of the exhaust gases can be decreased in low-speed (and partly medium speed) marine diesel engines to a level 250–300 °C; under partial loading conditions, it is significantly lower, which can complicate energy regeneration in waste heat utilization boilers where water is used as the working fluid [17,18].

Compared to water, organic working fluids have a significantly lower boiling point, and they do not suffer from the aforementioned disadvantages; simultaneously, these fluids can considerably extend the temperature range of the cogeneration cycle. This case creates conditions for increasing the energy efficiency indicators. Although this technology is widely used in onshore plants, research investigating the implementation of cogeneration cycles for maritime transport for practical applications remains lacking [19]. The aforementioned aspects are related to cycle energy efficiency indicators of the power plant in a wide operating load range, and it is necessary to select a rational strategy for managing the operational indicators of the cycle (e.g., the flow characteristics and indicators of the regeneration forms of energy in the power units of the cycle) with the load modes of the main power plant and the effect of employing a strategy to realize the working fluid supply characteristics and effect of cycle realization based on external conditions. Further, the technological substantiation of the working fluid type to the greatest extent is important, and it meets the requirements for achieving high energy efficiency and operational indicators of the cycle [20]. The most common technological solutions include typical gas turbine engines used to generate mechanical power through the open and closed Brayton cycle, organic Rankine cycle (ORC) (a closed loop thermodynamic operating cycle where the working fluid is constantly evaporated and condensed), and the Kalina cycle-modified Rankine cycle (where a mixture of fluids, i.e., ammonia-water, is used as the working fluid) [20–22].

The most significant factor for selecting the cycle for waste heat utilization systems is the source heat temperature. Suitable temperature ranges to achieve the optimal efficiency for the Brayton cycle, Kalina cycle, and ORC are 800 °C, between 10–450 °C, and 90–300 °C [23]. Kaiko et al. compared ORC and Brayton cycles for marine applications and found that the Brayton power output increased more at high exhaust gas temperatures compared to that for ORC; they found that ORC is better for power generation at temperatures up to 680 °C, whereas the Brayton cycle is better at higher temperatures, which makes it less attractive for marine applications [24]. As the efficient temperature ranges of the Kalina and Rankine cycles are similar, Bombarda et al. [25] stated that both cycles produce equal amounts of power output in marine diesel engines; however, the Kalina cycle requires very high maximum pressure for high thermodynamic performance and expensive no-corrosion materials, such as a water-ammonia working fluid. More detailed research on the Kalina cycle for marine applications is currently ongoing [25,26].

The ORC has the following advantages over the Brayton and Kalina cycles for marine applications: high flexibility, safety, low maintenance requirements, and good thermal performance. The ORC makes it possible to realize energy recovery from a low-temperature heat source [27–29]. For the ORC, organic refrigerants or hydrocarbon compounds are used as working fluids because of their significantly lower boiling point than water, which results in a lower input requirement for producing power [27,30,31]; further, it expands

the temperature range of the working fluid cycle, which results in higher energy efficiency parameters. The ORC is a proven and reliable technology that can convert low-medium heat sources into useful power. System efficiency can be optimized by selecting a proper working fluid operated under suitable working conditions to achieve the maximum energy performance. Hung et al. [32] researched 11 ORC working fluids and their thermodynamic performance, and they found that the suitable working conditions of various fluids can be identified based on their saturation vapor curves and response to the temperature energy source.

Meanwhile, the ORC has been used to convert thermal energy from stationary energy sources for industrial purposes, for example, during the combustion of biomass, geothermal energy systems, and collected heat lost by industrial processes [33,34]. In recent decades, the growing importance of improving energy efficiency and reducing air pollution from vehicles has accelerated research into energy efficiency (WHR) technologies for ICEs. For example, a comparative assessment of the steam Rankine energy cycle (SRC) and ORC is provided for diesel engines based on 45 data points [34]. The main topics of research include the effect of the turboexpander operation on the efficiency of the device and the selection of the working fluid. The application of various WHR technologies to a marine two-stroke engine has been investigated in previous studies [19,35]. Further, review proposed ORCs as a promising technology [35]. Another study considered modern technologies for reducing greenhouse gas emissions from shipping, including the WHRS [36].

From a technological point of view, WHR technologies are more suitable for applications in marine systems because of their substantially stationary operating modes compared to those used in land transport systems and the large dimensions of marine engine rooms. The adoption of the directive on $CO_2$ emissions from ships [1] has provided a new impetus to the further development and improvement (SRC) of WHRS ships. Most studies, including reviews, focus on only certain aspects of the use of the ORC as an additive for most energy-efficient applications of ships.

Song et Al. examined waste heat recovery with ORC of 996 kW marine diesel engine and achieved results that rational system configuration is able to reach 10.2% power increase [37]. ORC is already used for marine application. In a service vessel with ORC, heat recovery systems showed that fuel saving can be achieved from 4 to 15%, which prompts quick payback time of the system [38,39].

One of the most comprehensive and extensive reviews on the application of WHRS in marine propulsion systems in terms of the aspects examined was conducted in Reference [38]. Studies by Swedish technical authorities and Swedish Department of Shipping summarized the experience of using WHR systems in recent decades from more than 180 scientific and technological sources. The authors substantiated the advantages and evaluated the possibility of using an organic single-stage blade-type detander ORC in three vessels: container vessels, bulk carriers, and oil tankers. This research analyzed alternative cycle structures, working fluids, cycle strategy components, controllers, and economic issues related to the profitability of ORC use. Based on the analysis of the fleet structure of the controlled ship control systems, the authors justified the use of WHRS for low-speed two-stroke marine engines in estimating the exhaust heat, bonnet cooling systems, and charge air heat potential.

Survey materials [40] indicate that some aspects of the use of ORCs on ships are yet to be fully investigated and analyzed owing to the complexity and variety of studies performed. Compared to other test objects [40], the use of ORC in four-stroke engines, which competes quite successfully with the majority of the fleet that use two-stroke low-speed engines, is attractive because of the higher exhaust temperature and corresponding WHR energy potential (20–25% heat balance) compared to that of low-speed diesel engines (15–20%). The exhaust gas of one of the [40] test objects (MAN 6S80ME-C9 diesel engine) achieves 19% of heat combustion of the fuel, whereas this value is 23% for the medium-speed engines, which are employed in the author's work.

The assumption of replacing the actual load cycle structure of the power plant with its average operating value is suitable for estimating the full energy potential in the operation based on heat balance; however, this does not facilitate the development of a strategy for implementing ORC across the full engine load range. Further, the efficiency of the use of a turbine with a regulated design that allows achieving an optimal ratio of cycle parameters underestimates the structure of the operational load cycle and the effect of seawater in the water area on ORC parameters.

The choice of the working fluid is another key solution for ensuring the energy efficiency of the ORC. An analysis of the application of various working fluids at sea shows that there are no unambiguous universal solutions [41]. Andreasen et al. [42] proved that *R245fa* provides the highest net power compared to *R134a*, *R32*, and their mixtures. In [41], the authors reported similar results. Soffiato et al. [43] compared the operating fluids *R134a*, *R125*, *R236fa*, *R245ca*, *R245fa*, and *R227ea* in simple ORC turbine generators using engine cooling system heat and found that *R227ea* provides the maximum net power. Kalikatsarakis and Frangopoulos [44] tested 11 pure fluids and 9 blends for use in WHR marine engines and showed that the *R245fa*, *R245ca*, and *R365mfc* blends (50/50) work as optimal working fluids in terms of energy efficiency. Koroglu and Sogut [45] concluded the optimality of *R113* suitability for marine transport.

The data presented above allow us to assume that, along with the thermophysical characteristics, the efficiency of working fluids in a ship application depends on the structure of the load cycle of the power plant and the change in the ORC boundary conditions during operation to the same extent, which is in contrast to that in industrial applications. Thus, it is advisable to extend the assessment of the efficiency of working fluids for the entire engine load cycle considering changes in the boundary conditions (temperature of the outboard water, detander control strategies, etc.).

Thus, to increase the energy indicators of the cogeneration cycle for marine transport use, it is rational to expand the study of its energy indicators for the alternative use of various types of working fluids (Wet, Dry, and Isentropic) [46]. This helps assess the effects on the cycle indicators in terms of the operating conditions of the vessel, which include the secondary energy sources of the main power plant of the vessel in practical operational load modes, and on rational strategies for cogeneration cycle parameter management. Further, it helps assess the effect of external temperature conditions within the operation area of the vessel. As the studies found by authors lack coherence of ORC system application for marine diesel engines in specific load modes, authors decided to expand the existing studies with the ORC system rational operational strategies management for the specific load modes of the marine diesel engine and the turbogenerator technical structure. The tasks listed by the authors were examined in the study of comparative model studies of the ORC as part of a medium-speed four-stroke power engine.

The single-stage ORC configuration with a centrifugal turbine was selected based on reviewed studies and the experience of the authors in using WHR technology, which is characterized by the priority of simple and reliable operation in shipping. The choice of working fluids is determined by the safety requirements for their use and production prospects from the standpoint of environmental protection.

## 2. Methodological Aspects of the Research

The logical sequence of the comparative studies performed on a graphic farm is shown in Figure 1.

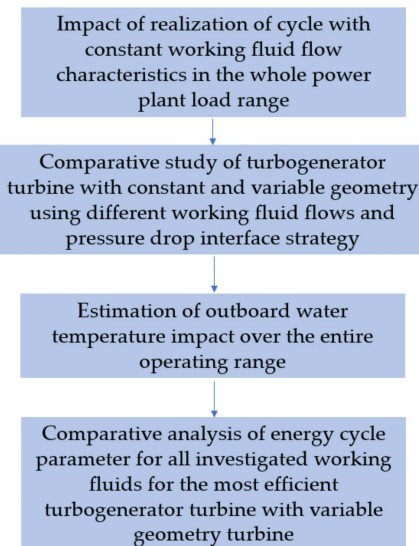

**Figure 1.** Logical sequence of comparative research plan.

The ORC scheme applied in these studies is presented in Figure 2. The thermodynamic cycle was simulated in Thermoflow, which is the leading simulation tool in the power and cogeneration industries. This software allows designing cogeneration cycles from the selected components with properties set by the users, and it can run the cycle simulation and obtain the results for the desired parameters (https://www.thermoflow.com, accessed on 1 July 2021).

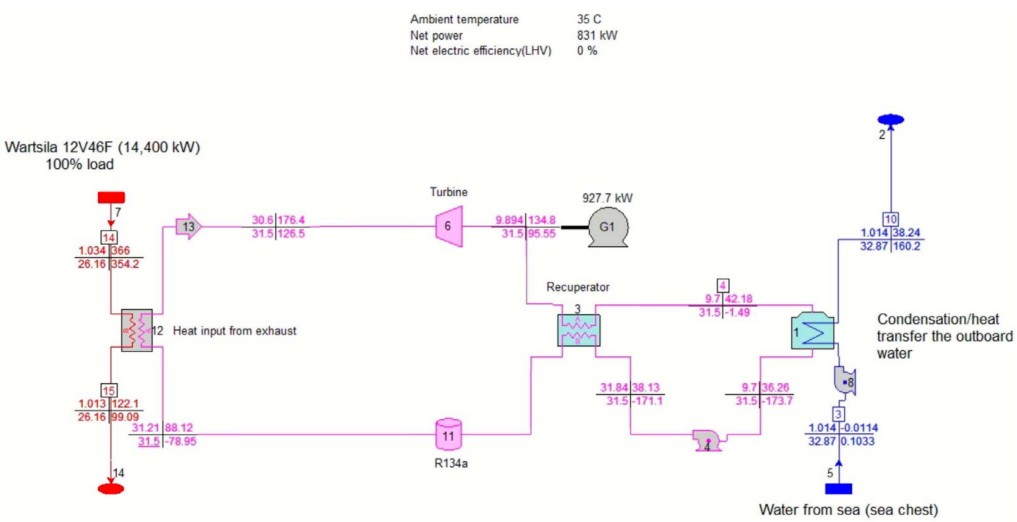

**Figure 2.** Single stage pressure organic Rankine cycle with recuperator: 1 = condenser; 2 = outboard; 3 = heat exchanger (recuperator); 4 = working fluid pump; 5 = sea chest inlet; 6 = turbine; 7 = exhaust gas inlet; 8 = sea water pump; 11 = working fluid tank; 14 = atmosphere.

In the Thermoflow software, a classic single-stage pressure ORC with a recuperator exchanger is designed to improve the energy efficiency indicators of the cycle.

The following principles are applied in terms of the components used in the cycle (Figure 2): the working fluid is heated and evaporated in a regenerative heat exchanger (pos. 3) to the saturated vapor state and the saturated vapor of the working material is supplied to the heat exchanger (pos. 12) in a turbogenerator turbine (pos. 6). The regeneration of the superheated working fluid vapor energy into mechanical work occurs from the condenser (pos. 1). The working fluid leaves in the state of the saturated liquid (by applying saturation condensation pressure according to the outboard water temperature).

The cogeneration cycle research was performed by combining its energy indicators and by applying it to a functioning 200-m-long ferry with a power plant of two medium-speed four-stroke main engines. The main specific engine parameters are listed in Table 1.

**Table 1.** Specifications of the medium-speed four-stroke main engine.

| | |
|---|---|
| Cylinder bore | 460 mm |
| Piston stroke | 580 mm |
| Cylinder output | 1200 kW/cyl |
| Number of cylinders | 12 |
| Speed | 600 rpm |
| Piston speed | 11.6 m/s |

The research was performed with the main diesel engine working at the specific test-type E3 cycle load modes of ISO 8178 (Table 2).

**Table 2.** Specifications for the medium-speed four-stroke parameters at the specific test-type E3 cycle load modes of ISO 8178.

| Load Modes | $P_e$, kW | $n$, rmp | $b_e$, g/kWh | $G_{air}$, kg/s | $G_f$, kg/s | $T_g$, °C |
|---|---|---|---|---|---|---|
| 100% | 1200 | 600 | 178.7 | 26.1 | 0.72 | 366 |
| 75% | 900 | 545 | 188.7 | 23.35 | 0.54 | 309 |
| 50% | 600 | 478 | 190.6 | 18.8 | 0.384 | 273 |
| 25% | 300 | 378 | 197.0 | 14.5 | 0.2 | 255 |

The working fluid selection in this research comprises *R134a*, *R141b*, *R142b*, *R123*, *R245fa*, and *isopentane*. They can be classified into three categories according to the slope of the saturated vapor curve shown in the T-S diagram. The Isentropic fluid has a vertical slope; the Dry fluids, a positive slope; and the Wet fluids, a negative slope. The choice of working fluids for this study is based on recent comparative studies because of the large variety of working fluids [29]. Previous studies have indicated that Isentropic fluids are considered the best fit. One of the authors' tasks is to evaluate this provision for a specific object of the research.

Two Isentropic, two Dry, and one Wet fluid are considered as the working fluids for the comparison. The fluids and their parameters are summarized in Table 3.

**Table 3.** Working fluid selection for research with main parameters.

| Working Fluid | Fluid Class | Molar Mass (kg/kmol) | Boiling Temp. at Atmospheric Pressure, °C | Critical Temperature, °C | Critical Pressure, MPa |
|---|---|---|---|---|---|
| R134a | Wet | 10.203 | −26.3 | 10.106 | 4059 |
| R141b | Isentropic | 11.695 | 32.05 | 20.435 | 4212 |
| R142b | Isentropic | 1005 | −9.12 | 13.711 | 4055 |
| R245fa | Dry | 13.405 | 15.14 | 15.401 | 3651 |
| Isopentane | Dry | 72.149 | 27.88 | 18.728 | 0.0338 |

The cycle was modelled according the boundary conditions. The energy indicators of the cogeneration cycle are evaluated by (1) differentially setting the flow of the working material in the system $G_{w.fl.}$ in the load modes of the power plant to achieve the maximum energy utilization of the exhaust gas (decreasing the exhaust gas temperature to the dew point 120 °C); (2) simulating a turbogenerator design with a fixed geometry turbine characterized by the hydrodynamic relationship between the working fluid flow $G_{w.fl.}$

and the degree of pressure drop in the turbine turbogenerator $\pi_T$; and (3) changing the fixed-geometry turbine design of the turbogenerator to a turbine with variable geometry (ensuring $\pi_T = const$ at different working material flows ($G_{w.fl.}$). The realization of the cycle at the external factors $-T_W$ outboard water temperature impact evaluation. Two values, 20 °C and 35 °C, are threshold values according to the registry (with prolongation of analytical evaluation to 0 °C) on the energy parameters.

The different organic working materials in the cogeneration Rankine cycle were evaluated according to the change in the energy efficiency index of the running engine and the effective COP, including the components, by evaluating the additional form $\delta\eta_{eR}$ of the mechanical energy generation power of the turbo-generator powerplant balance.

The cogeneration cycle COP structure includes the energy transformation efficiency parameters in characteristic cycle nodes.

$$\eta_{eR} = \eta_{h.ex} \cdot \eta_{tg.r} \cdot \eta_{T.ad} \cdot \eta_m \cdot \Psi. \tag{1}$$

$\delta\eta_{eR}$ is obtained from the compared solution $Q_{ex.g} = G_f \cdot H_u \cdot Q_{ex.g}$ and $P_e = \frac{G_f \cdot H_u \cdot \eta_e}{3600}$ as

$$\eta_{eR} = \frac{Q_{ex.g}}{\eta_e} \cdot \eta_{h.ex} \cdot \eta_{tg.r} \cdot \eta_{T.ad} \cdot \eta_m \cdot \Psi \tag{2}$$

where $Q_{ex.g}$, $\eta_{h.ex}$, $\eta_{tg.r}$, $\eta_{Tad}$, $\eta_m$, $\Psi$, $H_u$, $G_f$, and $\eta_e$ denote the relative part of the exhaust gas energy of the power plant (heat balance kJ/h), thermal COP of the exhaust gas heat exchanger, relative COP of the turbogenerator, turbogenerator internal (adiabatic) COP, turbogenerator mechanical COP, exhaust heat recovery factor, lower calorific value of fuel used by the traction engine (kJ/kg), hourly fuel consumption of the main engine (kg/h), and main power plant coefficient of performance.

The exhaust heat recovery factor is determined according to the ratio of the enthalpies of the exhaust gas before ($h_{tg1}$) and after ($h_{tg2}$) the heat exchanger and the enthalpies of the exhaust gas at dew point $h'_{tg2}$:

$$\Psi = \frac{h_{tg1} - h_{tg2}}{h_{tg1} - h'_{tg2}} \tag{3}$$

The thermal efficiency of the turbogenerator is determined by the ratio of the enthalpies of the working fluid before $(h_{tg1})$ and after $(h_{tg2})$ the turbine and the enthalpy of the working fluid $h'_{tg2}$ corresponding to the boiling temperature:

$$\eta_{tg.r} = \frac{h_{R1} - h_{R2}}{h_{R1} - h'_{R2}} \tag{4}$$

These fixed values are accepted in the calculations (fixed values according to existing models): $\eta_{Tad} = 0.7$; $\eta_m = 0.95$; $\eta_{h.ex\ (exhaust\ gas\ heat\ exchanger)} = 0.97$; $\eta_{h.ex\ (condenser)} = 0.97$; $\eta_{h.ex\ (rekuperator\ heat\ exchanger)} = 0.95$; pressure drop in exhaust gas heat exchanger 2%; and pressure drop in recuperator heat exchanger 2%.

In an analysis of the cogeneration cycle components, the turbine power generated by the turbo-generator $P_{gen}$ was determined in parallel by several analytical dependencies. $P_{gen}$ can be described in several forms to evaluate its possible improvement for identifying the factors that determine the efficiency of the cogeneration cycle. On the other hand, it allows determining the relationship between the turbogenerator operating parameters for their reasonable choice:

$$P_{gen} = \frac{G_{w.fl.}\ (h_{tg1} - h_{tg2})}{3600} \tag{5}$$

where: $G_{w.fl.}$ denotes the working material flow (kg/h); $h_{tg1}$ and $h_{tg2}$ represent the working material enthalpy before and behind the turbo-generator turbine (kJ/kg).

Then, the total mechanical energy generated by the main engine (ME) with the COP of the turbogenerator mechanical energy is calculated as:

$$\eta_{\Sigma e} = \frac{(P_e + P_{tg})3600}{H_u \cdot G_f}, \text{ and its change}$$

$$\delta\eta_{\Sigma e} = \frac{\eta_{\Sigma e} - \eta_e}{\eta_e} = \frac{(P_e + P_{tg})3600}{H_u \cdot G_f} - \frac{(P_e)\cdot 3600}{H_u \cdot G_f} = \frac{P_{tg}}{P_e} \qquad (6)$$

The efficiency of the cogeneration cycle in terms of power according to the energy efficiency appliance, COP, of the turbogenerator is described by:

$$P_{gen} = \frac{Q_{ex.g} \cdot \eta_{h.ex\ (exhaust\ gas)} \cdot \eta_{tg.r} \cdot \eta_{T.ad} \cdot \eta_m \cdot \Psi}{3600}. \qquad (7)$$

According to Equation (2), $\delta\eta_{eR}$ is obtained from the compared solution $Q_{ex.g} = G_f \cdot H_u \cdot q_{ex.g}$, and $P_e = \frac{G_f \cdot H_u \cdot \eta_e}{3600}$. We obtain a function from the relative values as:

$$\delta\eta_{\Sigma e} = \frac{Q_{ex.g}}{\eta_e} \cdot \eta_{h.ex} \cdot \eta_{tg.r} \cdot \eta_{T.ad} \cdot \eta_m \cdot \Psi \qquad (8)$$

When performing comparative studies on the prediction of cycle efficiency, the value of $G_{w.fl.}$ is antecedent as a constant with the main engine running at the rated nominal power, provided that the exhaust gas temperature outside the regenerative heat exchanger does not fall below the dew point. If there is a need to reduce $G_{w.fl.}$ in the part-load engine load modes because of the same condition, it is performed at the interface with $\pi_T$ changes.

When performing cogeneration cycle comparisons at the level when the turbo-generator type and its actual characteristics are not identified, the classical second equation of turbomachinery theory is used to determine the relationship between $G_{w.fl.}$ and $\pi_T$ [47,48].

Based on these classical turbomachinery equation, an equation is derived for the different load modes evaluation, which reveals the connection between the specific load mode, the working fluid flow of the system, and the pressure drop ratio of the turbogenerator.

$$\frac{G_{w.fl.\ (x)}}{G_{w.fl.\ (25)}} = \frac{\sqrt{\pi_{T(x)}^{K_R/2} - \pi_T^{\frac{K_R}{(K_R-1)}}}}{\sqrt{\pi_{T(25)}^{K_R/2} - \pi_{T(25)}^{\frac{K_R}{(K_R-1)}}}} \cdot \left(\frac{T_{1(25)}}{T_{1(x)}}\right)^{0.5} \qquad (9)$$

At the assumed $\pi_T = invar$ and the identified flow rates $G_{w.fl.\ (25)}$ and $G_{w.fl.\ (x)}$, the value of $\pi_{T(x)}$ is determined iteratively until the difference between the value of $\pi_{T(x)}$ and the values calculated according to Equation (9) before the last iteration does not exceed 2–3%.

## 3. Results

The comparative analysis of the cogeneration cycle implementation strategy is comprised of several aspects, including the regulation of the cycle working fluid flow according to the load mode of the main power plant and the cogeneration cycle turbo-generator turbine design with a fixed and variable geometry, respectively (design that allows the realization of different working fluids and pressure drop ratios in turbogenerator turbine interface strategies). Accordingly, the influence of external conditions, such as outboard water temperature, on energy efficiency indicators was evaluated.

### 3.1. Comparative Analysis of ORC Realization Strategy

### 3.1.1. Regeneration Cycle $(G_{w.fl.} - \pi_T) = const$ Strategy

The analysis starts with a relatively simple practical realization of the cogeneration cycle control variant, which is characterized by practical implementation: the turbogenerator

operates in a steady-state mode; the flow rate of the working fluid material and the pressure expansion of the turbogenerator in its turbine are constant under all load conditions of the power plant from 25% to 100% of the rated power, i.e., $G_{w.fl.} = 9.5 \frac{kg}{s}$; $\pi_T = 3.09$.

The maximum value of the working material flow is limited by the dew point $T_{ex.g.}$, which corresponds to the level of the exhaust gases of the power plant, leaving the heat exchanger (pos. 12, Figure 2). When the temperature drops below 120 °C, the conditions cause the condensation of sulfuric acid from sulfur oxides in the exhaust gas [49].

The limits for $G_{w.fl.}$ are set at the lowest values of the exhaust gas at 255 °C at a minimum load of 25% $P_{e\ nom}$ mode: $G_{w.fl.} = 4.6$–9.5 kg/s.

The value of the parameter $G_{d.sk}$ is not equalized for different working fluids because it is already limited, according to $T_{ex.g.}$.

In medium and high-load power plant modes, there is a large reserve of unused exhaust gas energy in the heat exchanger owing to the higher energy efficiency of the exhaust gas. The energy indicators of the cogeneration cycle are presented in Table 1.

Significant differences in the cogeneration cycle COP in the operating 25–100% $P_{e\ nom}$ range with different types of working fluids are not observed: Wet-type *R134a* working fluid COP $\eta_{cog.c.}$ accounts for 15–4.5%; Isentropic *R141b* and *R142b* account for 13.1% to 3.9% and 14.8% to 4.5%, respectively; and Dry liquid *R245fa* accounts for 13.1% to 3.9%.

The minimum $\eta_{cog.c.}$ values achieved at that time are also typical for the (Dry) *isopentane* liquid, i.e., 8.8–2.7%.

Maximum values of $\eta_{cog.c.}$ are common for the minimum load mode of 25% $P_{e\ nom}$, whereas the minimum values are common for the nominal rated power mode of $P_{e\ nom}$. The simulation results are presented in Table 4.

The trend changes in the $\eta_{cog.c.}$ in the power plant load modes is determined, to a large extent, by the exhaust energy potential utilization factor parameter Ψ.

The internal energy potential of the exhaust gas increases with an increase in the power plant load, and its utilization in the heat exchanger at $G_{w.fl.} = const$ decreases. Changes in the parameter Ψ values do not depend on the type of the working fluid and range from 0.99 to 25% $P_{e\ nom}$ and range to 0.30 for $P_{e\ nom}$.

The mechanical energy performance difference generated in the turbogenerator does not exceed ~10% (except for the Dry-type *R245fa*) among all evaluated variants for the working fluids. The Dry-type working fluids have a lower generated mechanical energy efficiency of 202–230 kW, whereas Isentropic-type fluids have 224–248 kW and the Wet-type have 250 kW.

Although a significant difference in $P_{gen}$ of 50 kW (*R134a* and *R245fa*) was achieved between the comparable variants, the differences in the total COP increase of the power plant in individual modes did not exceed 0.3–1.3% ($P_{e\ nom}$ and 25% $P_{e\ nom}$ modes, respectively). The functioning of the cogeneration cycle in terms of the generated mechanical energy increased the cogeneration cycle COP compared to that without cogeneration from 7.1–7.7% in the 25% power plant load mode to 2.2–2.3% in the 100% power load mode. No differences between the use of working fluids in the cogeneration cycle were observed (0.2–0.3%) when assessing the integrated values of COP during the entire operating cycle according to the conditions of ISO8178.

There is no significant difference for the practical application of different types of working fluids if the cogeneration cycle is functioning ($G_{w.fl.}$, $\pi_T$ = constant) when evaluating the energy effect in the operating power plant load modes of the cycle.

For practical reasons related to the reliability of the system, it is rational to choose the condensing pressure of working fluids close to that of atmospheric pressure.

According to this operational aspect, *R141b* and *isopentane* are alternatively preferred for use with condensation saturation pressures of 1.3–1.5 bar. The working fluid *R134a* is less rational as it is characterized by a pressure behind the turbine of the turbogenerator and a condenser of approximately 9.9 bar.

**Table 4.** Results of cycle simulation: strategy $G_{w.fl.} = const$.

| Working Fluid | Load Mode. % | Exhaust Gas Temperature, °C (poz. 12) | | Working Fluid Temperature, °C (poz. 6) | | Enthalphy of Working Fluid, kJ/kg (poz. 6) | | Flow, kg/s | Pressure, bar (poz 6) | | $\pi_T$, poz 6 | $P_{gen}$ kW | $\eta_e$ | $\eta_{\in(cog.c.)}$ | $\delta\eta_e$ | $\eta_{cycl.}$ | $\eta_{cog.c.}$ | $\eta_{tg.r}$ | $\eta_{cog.c.}$ | $P_{gen}$ | $\Psi$ |
|---|---|---|---|---|---|---|---|---|---|---|---|---|---|---|---|---|---|---|---|---|---|
| | | Before | After | Before | After | Before | After | System | Before | After | | | | | | | | | | | |
| R134a | 100 | 366 | 293.7 | 176.4 | 134.8 | 126.5 | 95.55 | 9.5 | 9.894 | 30.6 | 3.093 | 249.5 | 0.469 | 0.480 | 2.34% | | 0.0452 | 0.2345 | 100 | 100 | 0.2989 |
| | 75 | 309 | 227.1 | 176.4 | 134.8 | 126.5 | 95.55 | 9.5 | 9.894 | 30.6 | 3.093 | 249.5 | 0.459 | 0.472 | 2.86% | 0.46935 | 0.0663 | 0.2345 | 100 | 100 | 0.4381 |
| | 50 | 273 | 170.3 | 176.4 | 134.8 | 126.5 | 95.55 | 9.5 | 9.894 | 30.6 | 3.093 | 249.5 | 0.44 | 0.458 | 4.01% | | 0.1019 | 0.2345 | 100 | 100 | 0.6736 |
| | 25 | 255 | 120.9 | 176.4 | 134.8 | 126.5 | 95.55 | 9.5 | 9.894 | 30.6 | 3.093 | 249.5 | 0.425 | 0.458 | 7.68% | | 0.1503 | 0.2345 | 100 | 100 | 0.9938 |
| R141b | 100 | 366 | 293.5 | 218.5 | 182.8 | 395.8 | 363.2 | 7.4 | 1.295 | 4.005 | 3.093 | 223.6 | 0.469 | 0.479 | 2.16% | | 0.0393 | 0.2035 | 87 | 90 | 0.2997 |
| | 75 | 309 | 226.9 | 218.5 | 182.8 | 395.8 | 363.2 | 7.4 | 1.295 | 4.005 | 3.093 | 223.6 | 0.459 | 0.471 | 2.62% | 0.46792 | 0.0576 | 0.2035 | 87 | 90 | 0.4386 |
| | 50 | 273 | 170 | 218.5 | 182.8 | 395.8 | 363.2 | 7.4 | 1.295 | 4.005 | 3.093 | 223.6 | 0.44 | 0.456 | 3.65% | | 0.0887 | 0.2035 | 87 | 90 | 0.6755 |
| | 25 | 255 | 120.6 | 218.5 | 182.8 | 395.8 | 363.2 | 7.4 | 1.295 | 4.005 | 3.093 | 223.6 | 0.425 | 0.455 | 6.95% | | 0.1307 | 0.2035 | 87 | 90 | 0.9958 |
| R142b | 100 | 366 | 294 | 190.2 | 150.6 | 148.3 | 114.5 | 8.2 | 5.1 | 15.77 | 3.092 | 247.6 | 0.469 | 0.480 | 2.32% | | 0.0447 | 0.2328 | 99 | 99 | 0.2978 |
| | 75 | 309 | 227.5 | 190.2 | 150.6 | 148.3 | 114.5 | 8.2 | 5.1 | 15.77 | 3.092 | 247.6 | 0.459 | 0.472 | 2.84% | 0.46924 | 0.0654 | 0.2328 | 99 | 99 | 0.4355 |
| | 50 | 273 | 170.8 | 190.2 | 150.6 | 148.3 | 114.5 | 8.2 | 5.1 | 15.77 | 3.092 | 247.6 | 0.44 | 0.458 | 3.99% | | 0.1007 | 0.2328 | 99 | 99 | 0.6705 |
| | 25 | 255 | 121.5 | 190.2 | 150.6 | 148.3 | 114.5 | 8.2 | 5.1 | 15.77 | 3.092 | 247.6 | 0.425 | 0.457 | 7.62% | | 0.1485 | 0.2328 | 99 | 99 | 0.9890 |
| R245fa | 100 | 366 | 293.7 | 160.8 | 133.7 | 128.7 | 104.1 | 9 | 2.448 | 7.57 | 3.092 | 201.8 | 0.469 | 0.478 | 2.00% | | 0.0393 | 0.2037 | 87 | 81 | 0.2989 |
| | 75 | 309 | 227.1 | 160.8 | 133.7 | 128.7 | 104.1 | 9 | 2.448 | 7.57 | 3.092 | 201.8 | 0.459 | 0.470 | 2.41% | 0.46673 | 0.0575 | 0.2037 | 87 | 81 | 0.4375 |
| | 50 | 273 | 170.4 | 160.8 | 133.7 | 128.7 | 104.1 | 9 | 2.448 | 7.57 | 3.092 | 201.8 | 0.44 | 0.45473 | 3.35% | | 0.0884 | 0.2037 | 87 | 81 | 0.6730 |
| | 25 | 255 | 121 | 160.8 | 133.7 | 128.7 | 104.1 | 9 | 2.448 | 7.57 | 3.092 | 201.8 | 0.425 | 0.452 | 6.34% | | 0.1305 | 0.2037 | 87 | 81 | 0.9928 |
| Isopentane | 100 | 366 | 293.9 | 221.6 | 197.9 | 755.1 | 701.5 | 4.638 | 1.53 | 4.732 | 3.093 | 229.9 | 0.469 | 0.479 | 2.20% | | 0.0266 | 0.1380 | 59 | 92 | 0.2989 |
| | 75 | 309 | 227.3 | 221.5 | 197.9 | 755.1 | 701.5 | 4.638 | 1.53 | 4.732 | 3.093 | 229.9 | 0.459 | 0.471 | 2.68% | 0.46828 | 0.0389 | 0.1380 | 59 | 92 | 0.4375 |
| | 50 | 273 | 170.7 | 221.5 | 197.9 | 755.1 | 701.5 | 4.638 | 1.53 | 4.732 | 3.093 | 229.9 | 0.44 | 0.456 | 3.74% | | 0.0600 | 0.1380 | 59 | 92 | 0.6736 |
| | 25 | 255 | 121 | 221.6 | 197.9 | 755.1 | 701.5 | 4.638 | 1.53 | 4.732 | 3.093 | 230.2 | 0.425 | 0.455 | 7.14% | | 0.0884 | 0.1380 | 59 | 92 | 0.9931 |

### 3.1.2. Regeneration Cycle $G_{w.fl.}$ = *variable* Strategy

The increase in the flow rate of the working fluid and enthalpy of the outgoing working fluid from the heat exchanger (pos. 12, Figure 2) affects the increase in the turbogenerator energy efficiency. In practice, such an operation strategy of a cogeneration cycle is implemented with the help of a variable-geometry turbine and an engine control unit (ECU) with a cogeneration cycle operational function.

Further, it is possible to implement the principle of the ICE-applied stepwise inflation system by gradually changing $G_{w.fl.}$ = *invar.* in the identified sections of the life cycle.

According to the strategy $G_{w.fl.}$ = *var.* ($\pi_T$ = *const*) in all load power plant modes, the cogeneration cycle COP ($\eta_{cog.c.}$) is constant, unlike that in the $G_{w.fl}$ = *const* strategy (Table 2, Figure 1). When $G_{w.fl.}$ = *const* with increasing load cogeneration cycle, $\eta_{cog.c.}$ decreases from 8–15% to 2.7–4.5% $P_{e\ nom}$. When $G_{w.fl.}$ = *var.* at $\eta_{cog.c.}$ = *const*, the effect of the energy efficiency increases and reaches 70% for all types of working fluids with a close linear approximation of the power plant (Figure 3), which ensures the realization of $\Psi$ = 0.99 in all load modes. The simulation results are presented in Table 5. Comparison of strategies $(G_{w.fl.} - \pi_T)$ = *const* and $G_{w.fl.}$ = *variable* presented in Figures 3–5.

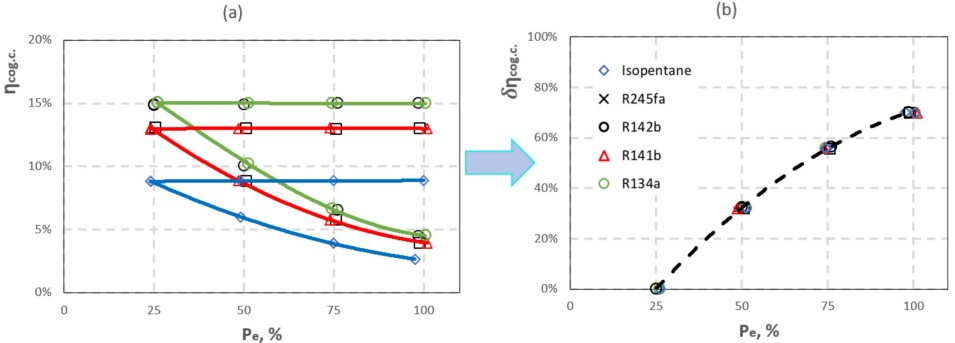

**Figure 3.** ORC (**a**) energy indicator $\eta_{cog.c.}$ comparison of strategies $G_{w.fl.}$ = *const* ( - - - ); $G_{w.fl.}$ = *var.* ( —— ); blue line = Dry fluid; red line = Isentropic/Dry fluids; green line = Wet/Isentropic fluids; (**b**) $\eta_{cog.c.}$ increase with strategy $G_{w.fl.}$ = *var.*

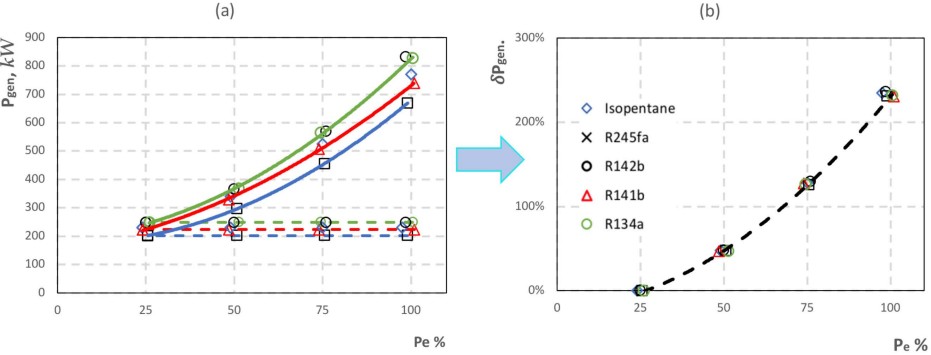

**Figure 4.** ORC (**a**) energy indicator $P_{gen}$ comparison of strategies $G_{w.fl.}$ = *const* ( - - - ); $G_{w.fl.}$ = *var.* ( —— ); blue line = Dry fluid; red line = Isentropic/Dry fluids; green line = Wet/Isentropic fluids; (**b**) $P_{gen}$ increase with cogeneration cycle with strategy $G_{w.fl.}$ = *var.*

**Table 5.** Results of cycle simulation: strategy $G_{w.fl.} = var$.

| Working Fluid | Load Mode. % | Exhaust Gas Temperature, °C (poz. 12) | | Working Fluid Temperature, °C (poz. 6) | | Enthalphy of Working Fluid, kJ/kg (poz. 6) | | Flow, kg/s | Pressure, bar (poz 6) | | $\pi_T$, poz 6 | $P_{gen}$ kW | $\eta_e$ | $\eta_{\in(cog.c.)}$ | $\delta\eta_e$ | $\eta_{cycl.}$ | $\eta_{cog.c.}$ | $\eta_{tg.r}$ | $\eta_{cog.c.}$ | $P_{gen}$ | Ψ |
|---|---|---|---|---|---|---|---|---|---|---|---|---|---|---|---|---|---|---|---|---|---|
| | | Before | After | Before | After | Before | After | System | Before | After | | | | | | | | | | | |
| R134a | 100 | 366 | 122.1 | 176.4 | 134.8 | 126.45 | 95.55 | 31.5 | 30.6 | 9.894 | 3.093 | 828.5 | 0.469 | 0.499 | 6.38% | 0.4810 | 0.1500 | 0.2345 | 100 | 99 | 0.9917 |
| | 75 | 309 | 121.8 | 176.4 | 134.8 | 126.5 | 95.55 | 21.5 | 30.6 | 9.894 | 3.093 | 565.1 | 0.459 | 0.486 | 5.80% | | 0.1498 | 0.2345 | 100 | 99 | 0.9905 |
| | 50 | 309 | 121.1 | 176.4 | 134.8 | 126.5 | 95.55 | 14 | 30.6 | 9.894 | 3.093 | 367.7 | 0.44 | 0.465 | 5.66% | | 0.1502 | 0.2345 | 100 | 100 | 0.9931 |
| | 25 | 255 | 120.8 | 176.4 | 134.5 | 126.5 | 95.55 | 9.5 | 30.6 | 9.894 | 3.093 | 249.5 | 0.425 | 0.458 | 7.68% | | 0.1503 | 0.2345 | 100 | 100 | 0.9939 |
| R141b | 100 | 366 | 122.8 | 218.5 | 182.8 | 395.8 | 363.2 | 24.42 | 4.005 | 1.295 | 3.093 | 738.6 | 0.469 | 0.496 | 5.75% | 0.4783 | 0.1298 | 0.2035 | 87 | 89 | 0.9891 |
| | 75 | 309 | 120.8 | 218.5 | 182.8 | 395.8 | 363.2 | 16.8 | 4.005 | 1.295 | 3.093 | 508 | 0.459 | 0.483 | 5.27% | | 0.1307 | 0.2035 | 87 | 89 | 0.9958 |
| | 50 | 273 | 121.4 | 218.5 | 182.8 | 395.8 | 363.2 | 10.86 | 4.005 | 1.295 | 3.093 | 328.5 | 0.44 | 0.463 | 5.12% | | 0.1301 | 0.2035 | 87 | 89 | 0.9910 |
| | 25 | 255 | 121.7 | 218.5 | 182.8 | 395.8 | 363.2 | 7.34 | 4.005 | 1.295 | 3.093 | 222 | 0.425 | 0.454 | 6.91% | | 0.1297 | 0.2035 | 87 | 89 | 0.9880 |
| R142b | 100 | 366 | 120.5 | 190.2 | 150.5 | 148.3 | 114.4 | 27.5 | 15.77 | 5.1 | 3.092 | 831.9 | 0.469 | 0.499 | 6.41% | 0.4810 | 0.1499 | 0.2328 | 99 | 100 | 0.9982 |
| | 75 | 309 | 120 | 190.2 | 150.5 | 148.3 | 114.4 | 18.83 | 15.77 | 5.1 | 3.092 | 569 | 0.459 | 0.486 | 5.83% | | 0.1502 | 0.2328 | 99 | 100 | 1.0003 |
| | 50 | 273 | 121.4 | 190.2 | 150.5 | 148.3 | 114.4 | 12.12 | 15.77 | 5.1 | 3.092 | 366.3 | 0.44 | 0.465 | 5.65% | | 0.1488 | 0.2328 | 99 | 99 | 0.9912 |
| | 25 | 255 | 121.3 | 190.2 | 150.5 | 148.3 | 114.4 | 8.21 | 15.77 | 5.1 | 3.092 | 248.1 | 0.425 | 0.457 | 7.64% | | 0.1487 | 0.2328 | 99 | 99 | 0.9903 |
| R245fa | 100 | 366 | 122.9 | 160.8 | 133.7 | 128.7 | 104.1 | 29.77 | 7.57 | 2.448 | 3.092 | 668.6 | 0.469 | 0.494 | 5.27% | 0.4761 | 0.1299 | 0.2037 | 87 | 80 | 0.9884 |
| | 75 | 309 | 122.6 | 160.8 | 133.7 | 128.7 | 104.1 | 20.28 | 7.57 | 2.448 | 3.092 | 455.4 | 0.459 | 0.481 | 4.78% | | 0.1295 | 0.2037 | 87 | 80 | 0.9854 |
| | 50 | 273 | 121.7 | 160.8 | 133.7 | 128.7 | 104.1 | 13.22 | 7.57 | 2.448 | 3.092 | 296.9 | 0.44 | 0.461 | 4.68% | | 0.1300 | 0.2037 | 87 | 81 | 0.9889 |
| | 25 | 255 | 121.2 | 160.8 | 133.7 | 128.7 | 104.1 | 8.985 | 7.57 | 2.448 | 3.092 | 201.7 | 0.425 | 0.452 | 6.34% | | 0.1303 | 0.2037 | 87 | 81 | 0.9913 |
| Isopentane | 100 | 366 | 120.5 | 221.6 | 197.9 | 755.1 | 701.4 | 15.52 | 4.732 | 1.53 | 3.093 | 769.9 | 0.469 | 0.497 | 5.97% | 0.4791 | 0.0888 | 0.1380 | 59 | 93 | 0.9979 |
| | 75 | 309 | 120.9 | 221.6 | 197.9 | 755.1 | 701.4 | 10.58 | 4.732 | 1.53 | 3.093 | 524.5 | 0.459 | 0.484 | 5.42% | | 0.0886 | 0.1380 | 59 | 92 | 0.9952 |
| | 50 | 273 | 121.1 | 221.6 | 197.9 | 755.1 | 701.4 | 6.852 | 4.732 | 1.53 | 3.093 | 339.7 | 0.44 | 0.463 | 5.27% | | 0.0883 | 0.1380 | 59 | 93 | 0.9924 |
| | 25 | 255 | 121.3 | 221.6 | 197.9 | 755.1 | 701.4 | 4.638 | 4.732 | 1.53 | 3.093 | 229.9 | 0.425 | 0.455 | 7.13% | | 0.0882 | 0.1380 | 59 | 92 | 0.9908 |

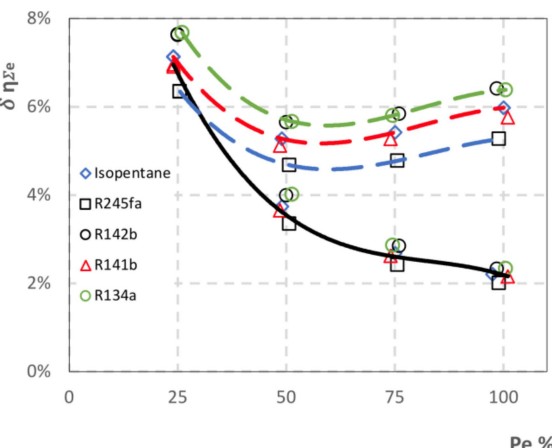

**Figure 5.** ORC realization strategy to influence the increase in ship power plant COP $\eta_{\Sigma e}$; $G_{w.fl.} = const$ ( $\mathbf{- - -}$ ); $G_{w.fl.} = var.$ ( $\mathbf{\longrightarrow}$ ); blue line: Dry fluid; red line: Isentropic/Dry fluids; and green line: Wet/Isentropic fluids.

Equal relative change in the $\eta_{cog.c.}$ parameter (Figure 3b) forms the basis to predict the expected effect for another working fluid based on the evaluation results of one working fluid; this includes converting the relative change in $\eta_{cog.c.}$ to absolute.

Compared with $G_{w.fl.} = const$ strategy, the energy efficiency ($P_{gen}$) of the turbogenerator increases from 470 to 570 kW (Figure 4a). The maximum increase in $P_{gen}$ is independent of the working fluid type, and the characteristics are equal for *R134a* (Wet), *R142b* (Isentropic), and *isopentane* (Dry) fluids. Interestingly, the relative changes for all evaluated working fluids remain the same, even with a different effect of $P_{gen}$ increase (Figure 4b).

The intensity of the increase in $P_{gen}$ increases with an increase in the power plant load regime in connection with the increase in the internal energy potential $Q_{ex.g.}$ of the exhaust gas.

According to the verification, the correlation coefficient between $P_{gen}$ and $Q_{ex.g.}$ is equal to 0.5 (determination ratio $R^2 = 0.98$).

The influence $\delta\eta_{\Sigma e}$ of the increase in the power plant COP (Figure 5) in the logical sequence achieved the largest $P_{gen}$ increase effect in the mode $P_{e\ nom}$, up to 3–4%.

In this context, the direction of the exhaust gas flow to the heat exchanger is controlled by the ECU, which is considered a cogeneration cycle energy efficiency operational tool that matches the energy parameters with the operating load of the power plant.

### 3.1.3. Influence of Outboard Water Temperature

The condensing conditions in the working fluid cogeneration cycle change according to the changes in the ship outboard water temperature $T_w$ changes as the ship is navigated (pos. 1). In terms of the energy efficiency and productivity of the cogeneration cycle, the changes in $T_w$ did not have a noticeable effect.

However, the change in $T_w$ results in changes in the condensation saturation temperature and pressure of the working fluid; the saturation temperature of the working fluid increases as $T_w$ increases; therefore, it is considered the necessary saturation pressure to ensure condensation. Thus, an increase in the working fluid pressure must be ensured in the branch of the cogeneration cycle from the turbogenerator to the circulation pump (pos. 4). In the branch of the high-pressure cogeneration cycle from the circulation pump to the turbo-generator, the pressure can be maintained in constant operation at different temperatures $T_w$.

However, it is rational to increase the working fluid pressure in the branch from the circulation pump to the turbogenerator proportionately to ensure $\pi_T$, $P_{gen}$, $\eta_{cog.c.}$ are constant in the power-plant load mode for maintaining the high-energy performance of the cogeneration cycle. Numerical modeling at $T_w = 20\ °C$ and $35\ °C$ confirmed the change in the energy cogeneration cycle parameters within a 1% error.

The increase in the pressure of the cogeneration cycle (caused by the increase in $T_w$) is characterized by a negative effect on the reliability of the cogeneration system, and it ensures the tightness of the system. Negative consequences can be caused by a decrease in $T_w$, which results in a saturation working fluid pressure below atmospheric pressure. The data presented in Table 6 indicates a decrease in the saturation pressure *R141b* at $T_w = 20$ °C.

**Table 6.** Impact of outboard sea water temperature on working fluid condensation pressure.

| $T_w$, °C | Working Fluid | | | | |
|---|---|---|---|---|---|
| | *R134a* | *R141b* | *R142b* | *R245fa* | *Isopentane* |
| 35 | 9.7 | 1.27 | 5.0 | 2.4 | 1.5 |
| 20 | 7.1 | 0.87 | 3.67 | 1.63 | 1.02 |
| 0 | 2.9 | 0.28 | 1.45 | 0.53 | 0.35 |

At $T_w = 0$ °C, close to the minimum $T_w$ level under the operating conditions, the atmospheric saturation pressure drops below the atmospheric pressure and in *R245fa* and *isopentane*.

If the atmospheric saturation pressure is lower, it tightens the requirements for the tightness of the cogeneration cycle system, similar to the case with overpressure. However, a leak with high pressure is related only to the escape of the cogeneration cycle from the system, which is dangerous from an environmental point of view. However, for practical reasons, the theoretically raised hypothesis about the heating of the overboard water that considers its real flow in the condenser is energetically unrealistic. For overboard water heating, the energy demand of 10 °C exceeds three times the energy potential of the power plant exhaust gas, for example, *R134a*: in the $P_{e\ nom}$ mode; it is ~20,000 kJ/s against 6670 kJ/s, and, in the 25% $P_{e\ nom}$ mode, it is 6000 kJ/s against 2000 kJ/s.

At leaks and pressures below atmospheric pressure, air enters the system from the atmosphere and degrades the energy efficiency and performance of the cogeneration cycle.

Thus, the choice of working fluid is emphasized not only to achieve better energy efficiency at the set energy efficiency parameters but also to ensure greater reliability of the cogeneration system.

### 3.1.4. Use of Variable Geometry Turbine of the Turbogenerator

The use of a turbogenerator turbine with a fixed geometry in a cogeneration cycle operating in a wide power plant operating range to regulate the cogeneration cycle is simpler than that of a variable geometry turbine.

The automatic adjustment of the $G_{w.fl.} - \pi_T$ parameter interface of the turbine turbogenerator occurs because of the changes in the power potential of the power plant exhaust gas in the variable load. For efficient cogeneration cycle operation, the turbine characteristics $\pi_T, \eta_{T.ad.} = f\left(G_{w.fl.}, \pi_T\right)$ are required to match the energy parameters of the power plant exhaust.

In numerical modeling, the analytical relationship $G_{w.fl.} - \pi_T$ is determined based on the classical theory of turbo machines in the second equation of a free turbogenerator [50,51].

Comparison of turbine construction configuration in ORC system with realization strategy is presented in Figure 6.

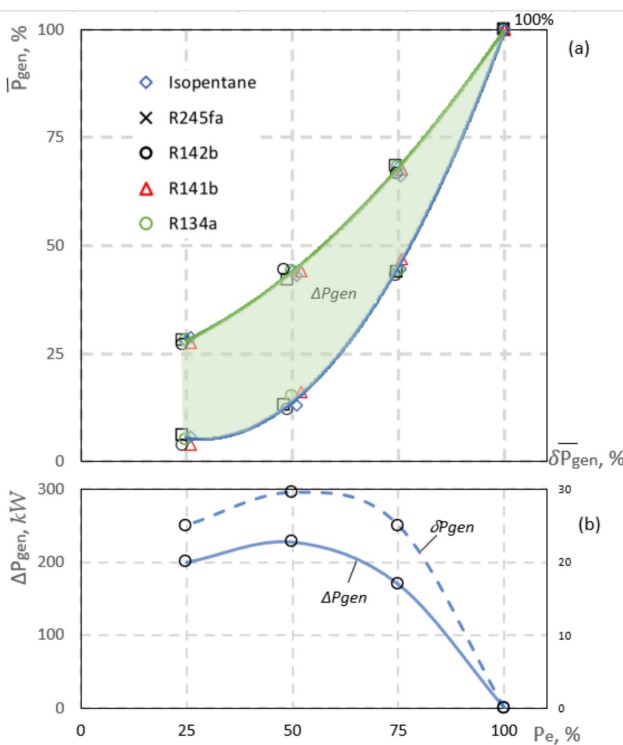

**Figure 6.** (**a**) $P_{gen}$ generated mechanical power in load cycle with ORC realization; blue line = $\pi_T$ = *variable*; green line = $\pi_T$ = *const.*; (**b**) turbine construction impact to $P_{gen}$ for comparative variants.

The obtained results are compared with the variable geometry turbine of the cogeneration cycle assembly turbogenerator.

In both cases, $G_{w.fl.}$ was selected for each power plant load mode in order to maximize the energy potential of the exhaust gas (reached $\Psi \sim 0.98 \div 0.99$).

For the variable geometry turbine, the value of $\pi_T$ was fixed at 3.09. The simulation results are presented in Table 7. The obtained results were compared with the variable-geometry turbine of the cogeneration cycle assembly turbo-generator. In both cases, $G_{w.fl.}$ was selected for each power plant load mode to maximize the energy potential of the exhaust gas (reached $\Psi \sim 0.98$–$0.99$). For the variable geometry turbine, the value of $\pi_T$ was fixed at 3.09; for a fixed geometry turbine, $\pi_T$ is determined by Equation (9) (power plant in the minimum 25% load $P_{e\ nom}$ mode assumes $\pi_T$ = 1.2). The energy efficiency of the turbogenerator was significantly reduced in the partial power plant load modes compared to that of the $\pi_T$ = *const* variable geometry turbine. Unlike the $\pi_T$ = *const* variant, the decrease in $P_{gen}$ is attributed to the decrease in both $G_{w.fl.}$ and $\pi_T$ in the part-load modes.

The maximum differences in the comparative turbine configurations of ~25–30% are observed in the medium and low load modes (Figure 6a). The calculated average load of the whole cogeneration cycle of the comparative variants was 65% of the turbogenerator of the variable geometry turbine and 40% of the fixed-geometry turbines because of equalizing the values of 100% $P_{gen}$ in relative terms. The energy efficiency loss of the cogeneration cycle was 20%. The differences in $P_{gen}$ ($\Delta P_{gen}$) in absolute power units are 230 kW or 30% of the nominal power of 820 kW (Figure 6b), and the average difference in the operating cycle is 160 kW.

The variable parameter $\pi_T$ also determines the decrease in turbocharger energy COP if $\eta_{tg.r}$ was constant in all load modes for a constant geometry turbine for the variable geometry turbine, and if it decreased from the maximum value to the $P_{e\ nom}$ mode to 6–7 times low load modes. The same range of change is typical for cogeneration cycle values. In parallel, the average COP of the cogeneration cycle decreases by 40–50%; for power plants with an integrated cogeneration system according to the ISO 8178 E3 cycle structure, the COP decreases by 2%.

**Table 7.** Results of cycle simulation: strategy $G_{w.fl.} - \pi_T$ (determined on the classical theory of turbo machines equation).

| Working Fluid | Load Mode. % | Exhaust Gas Temperature, °C (poz. 12) | | Working Fluid Temperature, °C (poz. 6) | | Enthalphy of Working Fluid, kJ/kg (poz. 6) | | Flow, kg/s | Pressure, bar (poz 6) | | $\pi_T$, poz 6 | $P_{gen}$ kW | $\eta_e$ | $\eta_{\in(cog.c.)}$ | $\delta\eta_e$ | $\eta_{cycl.}$ | $\eta_{cog.c.}$ | $\eta_{tg.r}$ | $\eta_{cog.c.}$ | $P_{gen}$ | $\Psi$ |
|---|---|---|---|---|---|---|---|---|---|---|---|---|---|---|---|---|---|---|---|---|---|
| | | Before | After | Before | After | Before | After | System | Before | After | | | | | | | | | | | |
| R134a | 100 | 366 | 122.1 | 176.4 | 134.8 | 126.5 | 95.55 | 31.5 | 9.7 | 30.6 | 3.15 | 828.5 | 0.469 | 0.499 | 6.38% | | 0.1500 | 0.2345 | 99 | 92 | 0.9917 |
| | 75 | 309 | 120.6 | 176.4 | 152.9 | 134.6 | 115.1 | 22.7 | 9.7 | 19.4 | 2 | 385.7 | 0.459 | 0.478 | 4.13% | 0.4715 | 0.0891 | 0.1385 | 96 | 100 | 0.9970 |
| | 50 | 273 | 120.4 | 176.5 | 165.3 | 138.6 | 128.7 | 15.4 | 9.7 | 13.78 | 1.42 | 133.5 | 0.44 | 0.451 | 2.39% | | 0.0437 | 0.0680 | 66 | 75 | 0.9974 |
| | 25 | 255 | 122.2 | 176.6 | 170.1 | 139.9 | 134 | 10.5 | 9.7 | 12 | 1.2 | 53.2 | 0.425 | 0.434 | 2.19% | | 0.0255 | 0.0402 | 100 | 100 | 0.9842 |
| R141b | 100 | 366 | 122.7 | 218.5 | 180.8 | 395.6 | 361.4 | 24.3 | 1.27 | 4.25 | 3.5 | 770.9 | 0.469 | 0.497 | 5.98% | | 0.1369 | 0.2144 | 90 | 86 | 0.9895 |
| | 75 | 309 | 120.8 | 218.8 | 196.8 | 397.3 | 376.7 | 17.5 | 1.27 | 2.6 | 2.05 | 334.2 | 0.459 | 0.476 | 3.65% | 0.4702 | 0.0817 | 0.1272 | 88 | 87 | 0.9956 |
| | 50 | 273 | 122.3 | 218.9 | 200.9 | 397.7 | 380.8 | 11.4 | 1.27 | 2.286 | 1.8 | 178.6 | 0.44 | 0.453 | 3.02% | | 0.0664 | 0.1044 | 100 | 100 | 0.9856 |
| | 25 | 255 | 121.4 | 219.2 | 213.7 | 398.6 | 393.4 | 8.1 | 1.27 | 1.54 | 1.2 | 38.95 | 0.425 | 0.433 | 1.79% | | 0.0206 | 0.0322 | 81 | 73 | 0.9900 |
| R142b | 100 | 366 | 120.1 | 190.2 | 150 | 148.1 | 113.8 | 27.5 | 5 | 16 | 3.2 | 896 | 0.469 | 0.501 | 6.85% | | 0.1517 | 0.2353 | 100 | 100 | 0.9995 |
| | 75 | 309 | 121.7 | 190.4 | 166.4 | 153.1 | 131.3 | 19.5 | 5 | 10.25 | 2.05 | 382.8 | 0.459 | 0.478 | 4.10% | 0.4715 | 0.0930 | 0.1454 | 100 | 99 | 0.9910 |
| | 50 | 273 | 122.7 | 190.6 | 180.6 | 156 | 146.5 | 13.2 | 5 | 6.85 | 1.37 | 112.6 | 0.44 | 0.449 | 2.10% | | 0.0395 | 0.0622 | 60 | 63 | 0.9841 |
| | 25 | 255 | 122.1 | 190.7 | 184.9 | 156.7 | 151.1 | 9.1 | 5 | 6.06 | 1.2 | 45.4 | 0.425 | 0.433 | 1.97% | | 0.0230 | 0.0363 | 90 | 85 | 0.9848 |
| R245fa | 100 | 366 | 122.9 | 160.8 | 130.8 | 127.8 | 101 | 29.5 | 2.4 | 8.45 | 3.52 | 720.1 | 0.469 | 0.495 | 5.63% | | 0.1426 | 0.2237 | 98 | 80 | 0.9885 |
| | 75 | 309 | 122.4 | 160.9 | 143.6 | 131.2 | 114.7 | 21 | 2.4 | 5.11 | 2.13 | 316.3 | 0.459 | 0.475 | 3.48% | 0.4687 | 0.0850 | 0.1335 | 91 | 82 | 0.9873 |
| | 50 | 273 | 123.4 | 161 | 152.8 | 132.8 | 124.7 | 14 | 2.4 | 3.5 | 1.46 | 104.4 | 0.44 | 0.4487 | 1.99% | | 0.0407 | 0.0646 | 61 | 59 | 0.9782 |
| | 25 | 255 | 122.5 | 161.1 | 157.1 | 133.4 | 129.5 | 9.7 | 2.4 | 2.91 | 1.2 | 35 | 0.425 | 0.432 | 1.68% | | 0.0201 | 0.0318 | 79 | 66 | 0.9817 |
| Isopentane | 100 | 366 | 120.5 | 221.5 | 195.9 | 754.3 | 696.5 | 15.4 | 1.5 | 5.175 | 3.45 | 820.3 | 0.469 | 0.499 | 6.33% | | 0.0955 | 0.1483 | 63 | 92 | 0.9982 |
| | 75 | 309 | 121.4 | 221.8 | 207.1 | 758.2 | 723.9 | 11 | 1.5 | 3.105 | 2.07 | 348.5 | 0.459 | 0.476 | 3.78% | 0.4702 | 0.0559 | 0.0873 | 60 | 90 | 0.9929 |
| | 50 | 273 | 121.7 | 221.9 | 214.8 | 760.1 | 743.2 | 7.4 | 1.5 | 2.16 | 1.44 | 115.6 | 0.44 | 0.449 | 2.14% | | 0.0273 | 0.0428 | 41 | 65 | 0.9892 |
| | 25 | 255 | 121.9 | 222 | 218.5 | 760.8 | 752.3 | 5.1 | 1.5 | 1.82 | 1.2 | 40.16 | 0.425 | 0.433 | 1.82% | | 0.0155 | 0.0244 | 61 | 76 | 0.9863 |

Therefore, within the limits of the performed numerical modeling, the highest energy efficiency and efficiency indicators were achieved for the turbogenerator assembly with a variable geometry turbine by combining the working fluid flow $G_{w.fl.}$ with the exhaust energy potential in all operating cycles to ensure the reliability of the cogeneration cycle operation when the condensation saturation pressure changes according to the outboard water temperature.

### 3.2. Discussion of Cogeneration Cycle Energy Efficiency Indicators

Not considering the relatively small losses in the condenser, water pump, and cogeneration heat exchanger, the cogeneration cycle energy COP is determined by the energy regeneration processes in the exhaust gas heat exchanger and the turbogenerator according to Equation (10).

$$\eta_{cog.c.} = \eta_{h.ex.(exh.gas\ heat\ exchanger)} \cdot \Psi \cdot \eta_{T.ad.} \cdot \eta_{Tm} \cdot \eta_{tg.r.} \tag{10}$$

The sequence of parameters in Equation (10) is determined by the structural and technological parameters of the units that ensure the quality of the heat transfer process, whereas the other parameters depend on the functional strategy that ensures the cogeneration cycle. The first parameters include the thermal COP of the heat exchanger, the adiabatic $\eta_{T.ad.}$, and the mechanical $\eta_m$ of the turbo-generator COP. This study is limited to the structural analysis of the cogeneration cycle; it does not evaluate the cycle composition in different models. Therefore, the values of the following parameters are assumed to be constant for all numerical modeling variants based on the widespread model data [52–54]:

$$\eta_{h.ex.(exh.gas\ heat\ exchanger)} = 0.95;\ \eta_{T.ad.} = 0.7;\ \eta_{T.m} = 0.95$$

Research evaluating energy performance improvement methods focuses on technologically regulated heat-efficiency parameters of the heat exchanger $\Psi$ and turbogenerator $\eta_{tg.r.}$, which are controlled technologically. Theoretically, the range of change for both parameters range from 0 to 1.

The parameter is $\Psi = 1.0$ in the case where the exhaust gas temperature of the heat exchanger reaches or is close to a predetermined dew point temperature margin. The exhaust gas temperature and flow rate vary over a wide range when the ship's main power plant operates under propulsion load conditions. Therefore, when the cogeneration cycle functions with a steady working fluid flow, $G_{w.fl.} = const$ in the maximum load mode of the parameter $\Psi$; this is achieved in one of the minimum load modes of the power plant performance. In one of the modes, the exhaust gas temperature falls below the dew point in the lower load range by regulating the $G_{w.fl.}$ at mid and high-load, and this results in the formation of sulfuric acid caused by the condensation. Therefore, the range of the functioning of the cogeneration cycle is narrowed.

In these studies, the regulation of $G_{w.fl.}$ in the low-load mode, i.e., 25% of the nominal (0.25% $P_{e\ nom}$), ensures $\Psi = 0.99$ in the case of using different working fluids. However, without changing $G_{w.fl.}$ in the higher load modes, $\Psi$ decreases appreciably, $P_{e\ nom}$ to 0.29–0.30, which automatically reduced the total cogeneration cycle $\eta_{cog.c.} \sim 70\%$ (Figure 3), regardless of the type of working fluid.

The individual control of $G_{w.fl.}$ in all power plant load modes ensures a constant value of $\Psi = 0.99$ and $\eta_{cog.c.}$, respectively. In parallel, $G_{w.fl.} = variable$ determines the energy efficiency of the turbo-generator as a result of which there is an increase in the COP of the integrated operating power plant with the cogeneration cycle from 2% (in the case of $G_{w.fl.} = const$ to 4–5%) was ensured.

Thus, the rational strategies for the operation of the cogeneration cycle include the adjustment of the circulating working fluid flow rate $G_{w.fl.}$ to the operating modes of the power plant to increase the energy efficiency and performance of the cogeneration cycle and to ensure the maximum $\Psi$ value.

This strategy is equally effective for turbogenerator design with variable and fixed turbine geometries to increase the energy efficiency of the cogeneration cycle.

The value of the parameter $\eta_{tg.r}$ is determined by the degree of pressure drop in the turbine $\pi_T$ and the temperature of the working fluid vapor before the turbine $T'_{w.fl.}$ because, after the turbine, the temperature $T''_{w.fl.}$ is limited by the saturation temperature $T'''_{w.fl.}$ (at corresponding pressures). The temperature $T_{w.fl.}$ value is ensured by regulating the flow of the working fluids and $T''_{w.fl.}$ temperature control (in order to $T''_{w.fl.} > T'''_{w.fl.}$) is the parameter $\pi_T$.

Another $\pi_T$ constraint on the expansion end pressure is determined by the relationship between the overboard water and working fluid condensation pressure in the condenser. In the construction of a cogeneration cycle without a regenerative heat exchanger (pos. 3, Figure 2), it is rational to increase the value of $\pi_T$ until the decrease in temperature $T''_{w.fl.}$ is close to $T''_{w.fl.}$. This is in parallel with the approach of $\eta_{tg.r.}$ to the maximum value of the turbogenerator energy efficiency (Equation (4)).

As in the investigated object, the $\pi_T$ increase is limited in the cogeneration cycle with a regenerative heat exchanger.

It is optimal to ensure the saturation temperature $T'''_{w.fl.}$ of the vapor of working fluids after the regenerative heat exchanger before entering the condenser. In turn, the heat exchange in a regenerative heat exchanger is determined by the need to convert the working fluid to saturated steam before entering the exhaust gas heat exchanger. The abandonment of the regenerative heat exchanger in the design is linked to the use of power plant exhaust energy for evaporating the working fluid, which in itself will limit $G_{w.fl.}$ and $T'_{w.fl.}$, thereby reducing the total cycle energy efficiency. In studies evaluating the cycle configuration, the parameter $\eta_{tg.r.}$ of most fluids reaches a close maximum level of 0.2–0.23 for a variable-geometry turbine (when $\pi_T = const$).

The flow rate $G_{w.fl.}$ and value of the $\pi_T$ parameter $\pi_T$ in a conventional fixed geometry turbine in the partial load conditions of the power plant decreased significantly, e.g., *R134a* and *R142b* from 0.24 to 0.035–0.04 at 0.25% $P_{e\ nom}$, analogously for *R141b* and *R245fa* decreased in the range of 0.022–0.025. At the same time, the COP of the cogeneration cycle of the investigated working fluids decreased from 0.15–0.10 to 0.015–0.020 in the working load mode $P_{e\ nom}$ (1.0–0.25).

Therefore, variable-geometry turbogenerator turbines provide $\pi_T = const$ over a wide power plant operating range, along with $G_{w.fl.} = invar$. The implementation strategy is characterized by incomparably higher energy parameters of the cogeneration cycle; however, it is structurally more complex.

A comparative evaluation of the energy parameters of the cogeneration cycle using different types of working fluids was performed ($\eta_{cog.c.} - P_{gen}$) in graphical form and presented in Figure 7.

Only the limited variation of the different working fluids in the studies can be conditionally evaluated by different species according to the efficiency of their use in the cogeneration cycle given configurations. In power plants and cycle load modes, the cycle has a higher energy performance when using Wet working fluids and partly Isentropic working fluids. *R134a* (Wet) is characterized by a high-energy cycle $\eta_{cog.c.}$ and $P_{gen}$ accepted in the evaluation as a 100% maximum.

Isentropic liquid *R142b* is used as efficiently in the cycle, and the energy parameters of the other Isentropic liquid *R141b* decrease to a maximum of 5–15%. Alternatively, Dry working fluids have either a lower energy efficiency or performance. The combination of cycle parameters ($\eta_{cog.c.} - P_{gen}$) with the working fluid *R245fa* deviates from the 100% maximum by 15–20% (i.e., reaches 85–80%). Another use of the Dry working fluid *isopentane* is characterized by a decrease in $P_{gen}$ by approximately 40% when there is a small deviation of approximately 10% from the 100% maximum according to the parameter $\eta_{cog.c.}$.

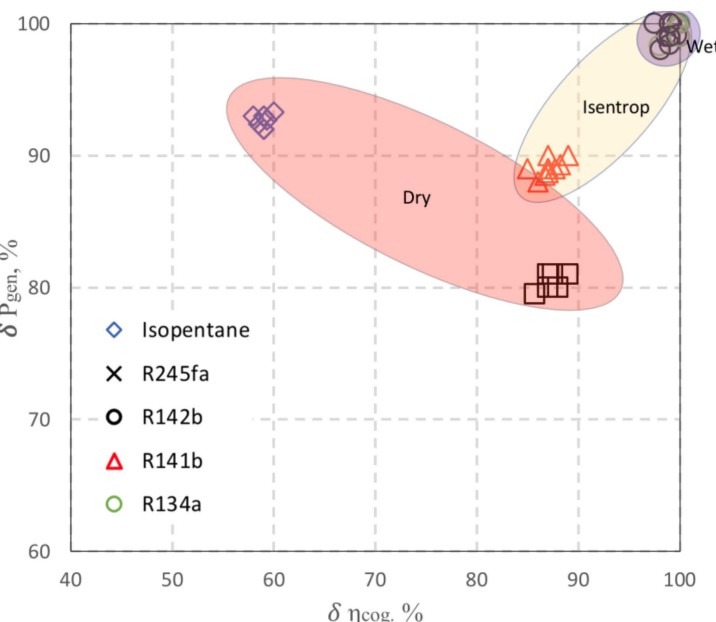

**Figure 7.** Comparative evaluation of working fluid type to energy indicators according ORC.

It is possible to speculate that the Dry-type working fluids are inferior for the ORC compared to other types based on the cogeneration cycle structural layout in the Mollier diagram field because of the less favorable enthalpy ratios in the superheated vapor and condensation zones (cooling and condensation line) for the cycle steps. However, this hypothesis requires more detailed research with different working fluids for practical conclusions.

## 4. Discussion

Without estimating the relatively small losses in the condenser, water pump, and regenerative heat exchanger, the cogeneration cycle energy COP is determined by the energy regeneration processes in the exhaust gas heat exchanger and the turbo-generator according to Equation (1).

The sequence of parameters in Equation (1) is determined by the structural and technological parameters of the units, which ensure the quality of the heat transfer process; the other parameters depend on the strategy employed to ensure the functioning of the cogeneration cycle. The first is the thermal COP of the exhaust gas heat exchanger and condenser, and the adiabatic $\eta_{T.ad.}$ and the mechanical $\eta_{T.m.}$ for turbine of turbogenerator COP.

This study is limited to the structural analysis of the cogeneration cycle without evaluating the cycle composition in different models. Therefore, the values of the following parameters, are assumed to be constant in all numerical modeling variants; the values are obtained based on data from widespread models [36,37]:

$$\eta_{h.ex.(exh.gas\ heat\ exchanger)} = 0.95;\ \eta_{T.ad} = 0.7;\ \eta_{T.m} = 0.95.$$

The focus of this research, which evaluates ways to improve the energy performance, is focused on the heat exchanger $\Psi$ and turbogenerator $\eta_{tg.r}$ energy efficiency parameters that are technologically manageable. Theoretically, the range of change in both parameters is from 0 to 1.

The exhaust heat recovery factor $\Psi = 1$, in the case when the exhaust gas temperature of the heat exchanger reaches or is close to the dew point with a certain predetermined margin.

The exhaust gas temperature and flow rate vary over a wide range when the ship's main power plant operates under propulsion load conditions. Therefore, when the cogen-

eration cycle functions with a steady working fluid flow, $G_{w.fl.} = const$ in the maximum mode of parameter $\Psi$, which is achieved in one of the minimum load modes of the power plant performance. The exhaust gas temperature falls below the dew point in the lower load range by regulating mid and high-load of the $G_{w.fl.}$ in one of the modes, which results in the formation of sulfuric acid because of condensation. As a result, the range of functioning of the cogeneration cycle is narrowed.

In these studies, for the regulation of $G_{w.fl.}$ in the low-load mode, 25% of the nominal (0.25% $P_{e\ nom}$) ensured $\Psi = 0.99$ when using different working fluids. However, without changing the $G_{w.fl.}$ in higher load modes, $\Psi$ reduced significantly, and $P_{e\ nom}$ to 0.29–0.30 level, which in itself reduces the cogeneration cycle $\eta_{cog.c.} \sim 70\%$ (Figure 3), regardless of the type of working fluid.

Individual control of $G_{w.fl.}$ in all power plant load modes ensures a constant value of $\Psi = 0.99$ and $\eta_{cog.c.}$, respectively. In parallel, $G_{w.fl.} = var.$ also determines the energy performance increase of the turbo-generator, which ensures the increase in the integrated operating power plant with the cogeneration cycle NVK from ~2% (in the case of ($G_{w.fl.} = const$) to 4–5%.

In summary, the rational strategies for the operation of the cogeneration cycle are to adjust the circulating working fluid flow $G_{w.fl.}$ to the power plant operating modes, ensuring a close maximum $\Psi$ value to increase the energy efficiency of the cogeneration cycle and energy efficiency.

This strategy for increasing the energy efficiency of a cogeneration cycle is equally effective for turbogenerator design with variable and fixed turbine geometries.

Relative COP value of turbogenerator $\eta_{turb.r.}$ is determined by the degree of pressure drop ratio ($\pi_T$) in the turbogenerator turbine and the temperature of the working fluid vapor before the turbine ($T'_{w.fl.}$) because the temperature after the turbine ($T''_{w.fl.}$) is limited by the saturation temperature, $T'''_{w.fl.}$ level. The highest possible temperature, $T'_{w.fl.}$, is ensured by regulating the flow of working fluids, and controlling the temperature of $T''_{w.fl.}$ (to $T''_{w.fl.} > T'''_{w.fl.}$) becomes $\pi_T$, which means that this parameter is also limited.

Another limitation of parameter $\pi_T$ caused by the expansion end pressure is determined by the pressure between the overboard water and the condensing pressure of the working material in the condenser. In a condensing cycle design without a regenerative heat exchanger (pos. 3, Figure 2), it is rational to increase the value of $\pi_T$ to a temperature drop of $T''_{w.fl.}$, which is close to $T'''_{w.fl.}$. The maximum value and energy efficiency of the turbogenerator were approached (Equation 6) in parallel with $\eta_{turb.r.}$.

In the cogeneration cycle with the regenerator heat exchanger, as in the investigated object, the increase in $\pi_T$ is limited by the heat exchange in the regenerative heat exchanger (pos. 3).

It would be optimal to provide $T'''_{w.fl.}$ for working liquid vapors with saturation temperature after the regenerative heat exchanger before entering the condenser.

In turn, the heat exchange in a regenerative heat exchanger is determined by the need to convert the working fluid before the exhaust gas heat exchanger to saturated steam. The disposition of the regenerative heat exchanger in the design is linked to the partial use of power plant exhaust energy to evaporate the working fluid, which will in itself limit the $G_{w.fl.}$ and $T'_{w.fl.}$ and, consequently, reduce the cycle energy efficiency.

In the performed tests using most working fluids, the relative COP of the turbogenerator $\eta_{tg.r}$ when evaluating the cycle configuration reached the level close to the maximum 0.2–0.23, with variable geometry turbines at $\pi_T = const$.

In a traditional fixed-geometry turbine design, the $G_{w.fl.}$ rate and the value of the $\pi_T$, relative COP of the turbogenerator $\eta_{tg.r}$, respectively, decreased with a decrease in the partial load conditions of the power plant.

For example, *R134a* and *R142b* range from 0.24 to 0.035–0.04 at the 25% $P_{e\ nom}$ power plant load, which is analogous to *R141b* and *R245fa* range from 0.22 to 0.25.

For the studied working fluids, the cogeneration cycle $\eta_{cog.c.}$ from 0.15–0.10 to 0.015–0.020 in the power plant load modes (100–25%) $P_{e\ nom}$.

Therefore, when using variable-geometry turbo-generator turbines which provide $\pi_T = const$ in a wide power plant operating range, together with $G_{w.fl.} = invar.$, the implementation strategy is characterized by incomparably higher energy parameters of the cogeneration cycle, although it is structurally more complex.

A comparative evaluation of the cogeneration cycle energy parameters using different working fluids was performed ($\eta_{cog.c.} - P_{gen}$) in the graphical form: energy efficiency–energy efficiency (Figure 5). The results of the cogeneration cycle test of the power plant in all load modes are presented, and the working fluids are identified in the figure according to the correspondence to a certain type.

However, it is possible to make relatively comparative assessments of different types of working fluids according to the achieved ORC efficiency indicators because of the limited number of different working material variants in the studies.

The cycle, which has previous energy performance, uses Wet working fluids and partially Isentropic working fluids. In most cases, it includes power plants and cycles in partial load modes, *R134a* (Wet) is characterized by the high values of both energy cycles $\eta_{cog.c.}$, and $P_{gen}$ parameters were assumed to be 100%. Isentropic *R142b* is not less efficient for use in the cycle, whereas the energy parameters of the other entropic *R141b* are lowered to a maximum of 5–15%. Dry working fluids have either a lower energy efficiency or lower energy performance. The combination of cycle parameters $\eta_{cog.c.} - P_{gen}$ with working fluid *R145fa* deviates from the 100% maximum by 15% and 20%, respectively (i.e., representing 85–80% of the Wet maximum).

The use of another Dry working material, *isopentane*, in a cycle is characterized by a decrease in $P_{gen}$ by approximately 40% with a small deviation of approximately 10% from the 100% peak in accordance with $\eta_{cog.c.}$ parameter.

Based on the structural layout of the cogeneration cycle in the Mollier diagram field [55,56], it can be speculated that the Dry parameters of the working material have a less favorable enthalpy proportion compared to the other types in the cycle section in superheated vapor and condensation areas (cooling and condensation line), from one side, and in the liquid heating and evaporation zones (preparation to get in the gas heat exchanger line), from the other side.

However, this hypothesis requires more detailed research on the basis of the energy and exergy balance, and this is planned for the future. The practical conclusions on the different types of working fluids in the cogeneration cycle for the attractiveness of a ship power plant will be provided with more in-depth studies of working fluid options.

## 5. Conclusions

Energy efficiency $\eta$ and performance ($P_{gen}$) studies have been performed on the more common one-stage Rankine cycle for ships while working on alternative Wet, Isentropic, and Dry organic working fluid types (*R134a*, *R141b*, *R142b*, *R245fa*, and *isopentane*). This research focuses on comparative studies of the cogeneration cycle energy parameters with the medium-speed four-stroke engine operating in a wide operational cycle load mode from 25% to 100% of the nominal power.

The comparative assessment of the cogeneration cycle fulfilling the parameters became operational strategies: working fluid flow and expansion pressure in turbine adjustment for a constant and variable geometry type of turbine, and for evaluating the influence of overboard water temperature.

The best indicators of cycle efficiency $\eta_{cog.c.}$ and generated mechanical energy $P_{gen}$ were obtained by implementing the strategy of the differentiated regulation of the working fluid flow control to maximize the energy potential of the utilization from the exhaust gas in different load modes and the implementation of the pressure drop rate $\pi_T = const$ in a variable geometry turbine model. The cogeneration cycle $\eta_{cog.c.}$ acquires the maximum

value in the entire load range for all working fluids: 15% (*R134a*), 8.8% (*Isopentane*), and the respective power plant COP and $\eta_{e\ plant}$ increased by 6.2% and 5.3%, respectively.

Insignificant differences in the energy parameters in practice are common for simpler cogeneration cycle implementation strategies with a fixed-geometry turbine of a turbo-generator with a self-change of working fluid according to the $G_{w.fl.} - \pi_T$ interface, and it provides $G_{w.fl.} = const$ (according to the exhaust energy potential in the low-load mode 25% $P_{e\ nom}$).

The influence of the change in the outboard water temperature ($T_w$) on the energy indicators of the cogeneration cycle in the temperature range 30–20 °C (and probably in the lower range) does not exceed ±1%.

Taking into account the limited amount of data, preliminarily, it can be stated that Wet (*R134a*) and Isentropic (*R141b*) fluids have better parameters, and, approximately, 10% lower energy efficiency cycle efficiency indicators are typical for Dry (*R145fa*, *isopentane*)-type working fluids; however, one of the Isentropic representatives, *R142b*, also has a similar decrease in the $\eta_{cog.c.}$ parameter.

**Author Contributions:** Conceptualization, S.L.; methodology, S.L.; software, T.Č.; validation, T.Č.; analysis, S.L. and T.Č.; investigation, S.L.; data curation, S.L. and T.Č.; writing—original draft preparation, S.L. and T.Č.; writing—review and editing, S.L. and T.Č.; visualization, T.Č.; funding acquisition, T.Č. All authors have read and agreed to the published version of the manuscript.

**Funding:** This research received no external funding. Funding provided by authors.

**Institutional Review Board Statement:** Not applicable.

**Informed Consent Statement:** Not applicable.

**Data Availability Statement:** Data sharing not applicable.

**Acknowledgments:** The authors are grateful to thermal engineering software "Thermoflow" developers for the opportunity to run cycle simulations for the experimental research.

**Conflicts of Interest:** The authors declare no conflict of interest as the parametric analysis is authors decision.

## Abbreviations

| | |
|---|---|
| $C_{P_R}$ | specific isobaric heat of the working fluid |
| $F_{T.ekv.}$ | the transverse increase of the turbine impeller and the guide apparatus equivalent to the passage area |
| $G_{w.fl.}$ | working fluid flow, kg/s |
| $G_f$ | hourly fuel consumption of the main engine, kg/h |
| $H_u$ | lower calorific value of fuel, kJ/kg |
| $K_R$ | polytropic indicator |
| $P_e$ | Power of the main power plant |
| $P_{e\ nom}$ | Nominal power of the main power plant |
| $P_{gen}$ | Power generated by turbogenerator, kW |
| $R$ | gas constant |
| $T_{RI}$ | working fluid temperature before turbine, °C |
| $T_w$ | outboard sea water temperature, °C |
| $Q_{ex.g}$ | relative part of the exhaust gas energy of power plant the heat balance kJ/h |
| $\pi_T$ | pressure drop ratio in the turbine |
| $T_{ex.g.}$ | exhaust gas temperature, K |
| $T_W$ | outboard water temperature, K |
| $\eta_{cog.c.}$ | cogeneration cycle COP |
| $\eta_{cycl.}$ | power plant cycle COP including generated energy $P_{gen}$ in ISO8178 (E3) operating test cycle |
| $\eta_e$ | main power plant coefficient of performance |

| | |
|---|---|
| $\eta_{\sum e}$ | total mechanical energy generated by ME with mechanical power of turbogenerator |
| $\eta_{h.ex}$ | thermal COP of the exhaust gas heat exchanger |
| $\eta_{tg.r}$ | the relative COP of the turbogenerator |
| $\eta_{T.ad}$ | turbogenerator internal (adiabatic) COP |
| $\eta_m$ | turbogenerator mechanical COP |
| $\Psi$ | exhaust heat recovery factor |
| $\Psi_T$ | outflow function |
| $\beta$ | impulse energy input coefficient |
| $\alpha$ | working fluid flow rate per pulse |

Abbreviations

| | |
|---|---|
| COP | Effective coefficient of performance |
| EEDI | Energy efficiency design index |
| ECU | Engine control unit |
| ORC | Organic Rankine cycle |
| SRC | Steam Rankine cycle |
| ICE | internal combustion engine |
| WHRS | waste heat recovery system |

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
