# Peer review of "Research of Organic Rankine Cycle Energy Characteristics at Operating Modes of Marine Diesel Engine"

_jmse, doi:10.3390/jmse9101049_

Round 1
Reviewer 1 Report
All the comments/issues have been addressed properly, and the paper can be published at its present state accordingly. Thank you.
Author Response
Dear Reviewer,
Thank you for your postive evaluation of our manuscript. We have revised our manuscript one more time according other reviewer comments Please find attached doc file for more information.

Reviewer 2 Report
Accepted.
In table 1 "Number of cylinder quantity" can be replaced by "Number of cylinder"
Author Response
Dear Reviewer,
Thank you for your postive evaluation of our manuscript. We have revised our manuscript one more time according other reviewer comments Please find attached PDF for more information.

Reviewer 3 Report
The manuscript deals with an interesting analysis of the energy characteristics of the ORC cogeneration operating modes of the marine diesel engine. The authors should improve the readability and scientific soundness, please carefully revise it to improve the quality, the reviewer has been highlighted several open points below:
Title, please summarize the title is too long, a maximum of 10-15 words could be sufficient.
Abbreviation section: please carefully revise the list, reporting the missing abbreviation and then reporting in alphabetic order.
Introduction section:
The authors could extend the introduction discussion reporting that advanced combustion concepts such as Dual Fuel, Partially Premixed combustion combined with alternative fuels (Natural Gas, Ethanol-diesel blends, etc.) could give a potential boost to the CI engine fuel economy and engine-out emissions reduction. However, they can highlight that the research and industry are working on fuel design and Advanced combustion concepts to improve the CO2 and improving the NOx-Soot trade-offs. The reviewer's suggestion is to take a look at the work performed at the Istituto Motori (look for Dr. Di Blasio) or Sandia (look for Dr. Mueller). It is worth mentioning their contribution. They provide a wider overview of the combustion technologies that have been tested able to improve the CO2 and improving the NOx-Soot trade-offs such as specific bowl design, innovative fuel injection systems.
Please, revise possible grammar or format mistakes (I have seen a green comma and a few words with the incorrect use of singular and plural).
Please emphasize the conclusion section summarizing the main outcomes in a short section. In general, the manuscript is too long, consider summarizing some parts.
I would like to recommend it for publication after a revision, please consider to highlights the novelty of this manuscript.
Author Response
Dear Reviewer,
The Authors reply is in the attached pdf file.

Round 2
Reviewer 3 Report
I would like to recommed it for publication.
This manuscript is a resubmission of an earlier submission. The following is a list of the peer review reports and author responses from that submission.
Round 1
Reviewer 1 Report
The paper studies the energy characteristics of the ORC cycle at operating modes of a big bore diesel engine.
The paper is well organized, well structured and deliberately discussed about the objectives and the scopes raised by the research methodology. In general, the paper can be published as it is.
Here are some recommendations that I believe they can improve the paper, but are not necessary.
- The tile could have been more scientific if there would not be any particular/trade mark name (Wärtsilä), just mention the range/category of the engine
- The introduction section could have been more worth if there would be more paper cited/discussed regarding the prior art/literature review.
- The conclusion section could have been more concise and understandable if you may make it a bit shorter. Making it in some bullet point format might be a choice.
Reviewer 2 Report
The paper provides a study as a way to reduce the amount of CO2 emissions by considering WHR systems in marine applications.
Few comments need to be considered:
1. The authors applied ORC to the marine systems using different types of fluid, however, they did not provide a detailed literature review and what is the novelty of this paper by considering that this single system with these fluids is used in other non-marine applications.
2. The authors mentioned that they do optimization procedures, what kind of optimization methods are used or it is just by performing sensitivity analysis?
3. In Table 1 Cylinder quantity can be changed to number of cylinder.
4. Do the authors make calibration of the numerical model and how they did it? is there a previously published paper for that?
5. Why did the authors consider the single-stage WHR and not the multi-stages?